# Feline Morbillivirus: Clinical Relevance of a Widespread Endemic Viral Infection of Cats

**DOI:** 10.3390/v15102087

**Published:** 2023-10-13

**Authors:** Maria Grazia Pennisi, Sándor Belák, Séverine Tasker, Diane D. Addie, Corine Boucraut-Baralon, Herman Egberink, Tadeusz Frymus, Katrin Hartmann, Regina Hofmann-Lehmann, Albert Lloret, Fulvio Marsilio, Etienne Thiry, Uwe Truyen, Karin Möstl, Margaret J. Hosie

**Affiliations:** 1Department of Veterinary Sciences, University of Messina, 98168 Messina, Italy; 2Department of Biomedical Sciences and Veterinary Public Health (BVF), Swedish University of Agricultural Sciences (SLU), P.O. Box 7036, 750 07 Uppsala, Sweden; sandor.belak@slu.se; 3Bristol Veterinary School, University of Bristol, Bristol BS40 5DU, UK; s.tasker@bristol.ac.uk; 4Linnaeus Veterinary Limited, Shirley, Solihull B90 4BN, UK; 5Independent Researcher, 64000 Pyrénées Aquitaine, France; draddie@catvirus.com; 6Scanelis Laboratory, 31770 Colomiers, France; corine.boucraut@scanelis.com; 7Department of Biomolecular Health Sciences, Faculty of Veterinary Medicine, University of Utrecht, 3584 CL Utrecht, The Netherlands; h.f.egberink@uu.nl; 8Department of Small Animal Diseases with Clinic, Institute of Veterinary Medicine, Warsaw University of Life Sciences—SGWW, 02-787 Warsaw, Poland; tadeusz_frymus@sggw.edu.pl; 9LMU Small Animal Clinic, Centre for Clinical Veterinary Medicine, LMU Munich, 80539 Munich, Germany; hartmann@lmu.de; 10Clinical Laboratory, Department of Clinical Diagnostics and Services, Vetsuisse Faculty, University of Zurich, 8057 Zurich, Switzerland; rhofmann@vetclinics.uzh.ch; 11Fundació Hospital Clínic Veterinari, Universitat Autònoma de Barcelona, Bellaterra, 08193 Barcelona, Spain; albert.lloret@uab.es; 12Faculty of Veterinary Medicine, Università degli Studi di Teramo, 64100 Teramo, Italy; fmarsilio@unite.it; 13Veterinary Virology and Animal Viral Diseases, Department of Infectious and Parasitic Diseases, FARAH Research Centre, Faculty of Veterinary Medicine, Liège University, B-4000 Liège, Belgium; etienne.thiry@ulg.ac.be; 14Institute of Animal Hygiene and Veterinary Public Health, University of Leipzig, 04103 Leipzig, Germany; truyen@vetmed.uni-leipzig.de; 15Institute of Virology, Department for Pathobiology, University of Veterinary Medicine, 1210 Vienna, Austria; karinmoestl@gmail.com; 16MRC-University of Glasgow Centre for Virus Research, Garscube Estate, Glasgow G61 1QH, UK; margaret.hosie@glasgow.ac.uk

**Keywords:** FeMV, cat, urine, CKD, TIN, host spectrum, epidemiology, experimental infection, pathology, diagnosis

## Abstract

Feline morbillivirus (FeMV) was first isolated in 2012 from stray cats in Hong Kong. It has been found in association with tubulointerstitial nephritis (TIN), the most common cause of feline chronic kidney disease (CKD). However, viral host spectrum and virus tropism go beyond the domestic cat and kidney tissues. The viral genetic diversity of FeMV is extensive, but it is not known if this is clinically relevant. Urine and kidney tissues have been widely tested in attempts to confirm associations between FeMV infection and renal disease, but samples from both healthy and sick cats can test positive and some cross-sectional studies have not found associations between FeMV infection and CKD. There is also evidence for acute kidney injury following infection with FeMV. The results of prevalence studies differ greatly depending on the population tested and methodologies used for detection, but worldwide distribution of FeMV has been shown. Experimental studies have confirmed previous field observations that higher viral loads are present in the urine compared to other tissues, and renal TIN lesions associated with FeMV antigen have been demonstrated, alongside virus lymphotropism and viraemia-associated lymphopenia. Longitudinal field studies have revealed persistent viral shedding in urine, although infection can be cleared spontaneously.

## 1. Introduction

The *Morbillivirus* genus (family *Paramyxoviridae*, subfamily *Orthoparamyxovirinae*) comprises several well-known RNA viruses of humans and animals, including measles virus, canine distemper virus (CDV), rinderpest virus (RPV) (globally eradicated in 2011), peste des petits ruminants virus (PPRV), and two viruses affecting marine mammals, cetacean morbillivirus (CeMV) and phocine distemper virus (PDV) [1].

Scarce data are available for cats regarding paramyxovirus infections, with the exception of their susceptibility to the highly pathogenic, zoonotic paramyxoviruses belonging to the *Orthoparamyxovirinae* subfamily (namely Hendra virus and Nipah virus of the *Henipavirus* genus), which have not yet been reported in Europe [2]. A paramyxovirus-like agent was isolated in 1981 from a cat with demyelinating lesions in the central nervous system (CNS) and intracytoplasmic inclusion bodies in glial cells [3]. CDV infection has never been documented in domestic cats, although limited viral replication was observed in macrophages in experimentally infected cats [4]. However, CDV and related morbilliviruses have been found in naturally infected wild and captive large felids [5,6,7,8,9,10], and disease outbreaks associated with these viruses are a significant threat to wildlife conservation.

In 2012, the new paramyxovirus, feline morbillivirus (FeMV), was isolated from stray cats in Hong Kong [11]. FeMV antigen was detected in renal tubular cells and the lymph nodes of two cats with tubulointerstitial nephritis (TIN). Subsequently, several studies have examined the properties of FeMV and have developed diagnostic methods for the detection of FeMV, allowing for investigation of the epidemiology and pathogenicity of the infection, particularly in the urinary tract. In this review, members of the European Advisory Board on Cat Diseases (ABCD) present the current state of knowledge on FeMV infection in cats, focusing on its clinical significance.

## 2. Virus Properties and Host Spectrum

FeMV is an enveloped, single-stranded RNA virus, with six genes encoding six structural and two nonstructural proteins [12,13]. Three of the structural proteins (nucleocapsid N, phosphoprotein P, and protein L) are found in the nucleocapsid. A matrix protein (M) is located between the nucleocapsid and the envelope. The glycoproteins H and F interact with the host cell membrane and are responsible for viral host spectrum, tissue tropism, and pathogenesis [13,14]. It has been shown that FeMV is phylogenetically distinct from other morbilliviruses [15]. Genetic analysis has demonstrated the presence of two distinct genotypes of FeMV sharing a genomic nucleotide sequence identity of approximately 78.2% [16]. Genotype 1 (GT1) was first identified in 2012 in Hong Kong and is found worldwide in cats [11], with detection confirmed in all studies conducted. In Asia, beyond Hong Kong, FeMV-GT1 has been identified in Japan, Thailand, Malaysia, and China [17,18,19,20,21]. In Europe, FeMV-GT1 has been detected in Germany, Italy, and Turkey [22,23,24,25,26]. In the Americas, FeMV-GT1 has been reported in the USA and Brazil [27,28,29]. Genetic heterogeneity of FeMV-GT1 isolates was found, and phylogenetic analysis of 29 publicly available whole genome sequences suggested the existence of two clades of FeMV-GT1 [14]. One clade, containing three clusters, includes the GT1 isolates from China, Japan, Thailand, Germany, Italy, Brazil, and the USA. The second clade includes only the GT1 strains from Italy [14]. Genotype 2 (GT2) was identified in Germany in 2018 [16]. It is not known how important genetic diversity of FeMVs is in determining the clinical outcome of infections.

The thermal sensitivity and stability of FeMV has been investigated by incubating viral stocks at various temperatures and measuring the replication capacity of the treated viruses in vitro [30]. Viral infectivity was reduced by exposure to high temperatures with incubation at 70 °C inactivating FeMV in two minutes. In contrast, FeMV was stable at 4 °C, retaining infectivity for at least 12 days [30]. This stability at low temperatures may allow indirect transmission to susceptible individuals, but further studies considering viral drying on contaminated surfaces are required.

The host spectrum of FeMV has been studied in vitro by the evaluation of viral replication in cell lines derived from 13 different mammalian species (including humans). These studies revealed that only cell lines derived from cats and African green monkeys were permissive to FeMV replication [31]. The feline cell lines that supported FeMV-GT1 replication included renal, fibroblastic, lymphoid, and glial cells [31]. The feline cells that allowed in vitro replication by FeMV-GT2 included renal cells, epithelial lung cells, lymphocyte subsets, monocytes, and primary cells from the cerebrum and cerebellum [32]. Virus tropism for different cell types has also been studied using immunohistochemistry [33]. FeMV antigens were detected in inflammatory cells residing in the blood vessels of the kidney and brain, in respiratory epithelial cells, alveolar macrophages, and to a lesser extent, the CNS [33]. These data support that systemic infections can occur with FeMV and that clinically relevant genotype differences in tropism may exist. Recently, the cellular receptors involved in FeMV infection have been studied [34,35]. FeMV infection of immune and epithelial cells is mediated by the same cell receptors used by other morbilliviruses to attach to the viral haemagglutinin. The signalling lymphocyte activation molecule family member 1 (SLAMF1 or CD150) is a set of primary cell receptors for morbilliviruses expressed on subsets of immune cells [35]. SLAMF1 was the host cell entry receptor used by a US strain of FeMV-GT1 in vitro [34]. The amino acid sequences of SLAMs differ amongst mammalian species and are likely to influence the host spectrum of morbilliviruses. Human, canid, and feline SLAMF1 amino acid sequences are different. Although both canine and feline SLAMF1-expressing cells were permissive to FeMV replication, feline SLAM cells were more permissive, such that massive syncytium formation was observed in the feline SLAM cells [35].

Unlike other morbilliviruses, the mechanism for FeMV-induced cell-to-cell fusion depends on cathepsin, a protease that is expressed in infected cells [34]. This cathepsin dependence of FeMV is shared with the zoonotic henipaviruses that infect cats (i.e., the Nipah and Hendra viruses) [34]. The reduced availability of cathepsin on feline lymphocytes, compared to monocytes, may explain the less severe lymphodepletion observed with acute FeMV-GT1 infection in feline experimental models compared to CDV infection in ferret experimental models [34]. The epithelial cell receptor that binds the viral haemagglutinin protein (H glycoprotein) appears to be nectin-4, similar to other morbilliviruses [34]. In support of this, in the FeMV-GT1 study [34], the H protein amino acids that were conserved included all those important for the H receptor/nectin-4 binding and function [34]. Further investigations are needed to confirm the H glycoprotein/nectin-4 interaction in FeMV infection. Different cell receptors may be involved and favour the excretion of FeMV in urine compared to the respiratory tract.

In addition to domestic cats, the host spectrum of FeMV infection includes wild felids, such as the *Leopardus guigna* in Chile [36] and the *Panthera pardus* in Thailand [37]. Azotaemia and TIN have been reported in two black leopards with FeMV infection in Thailand; FeMV could be a threat for susceptible endangered host species [37].

Nasal and oral swabs from dogs with respiratory disease in Thailand were tested reverse transcriptase–polymerase chain reaction (RT-PCR) positive for FeMV RNA, and FeMV-GT1 was subsequently isolated from swabs and lung samples of a dead dog [38]. The FeMV-GT1 sequences obtained in this study showed 97.5–99.2% identity with sequences derived from domestic cats in Thailand, Hong Kong, and Japan. An FeMV prevalence of 12.4% (14/113) was found in dogs in Thailand, and six of the PCR-positive dogs were co-infected with other respiratory viruses (comprising canine corona-, canine herpesvirus, and/or CDV). Immunohistochemistry (IHC) confirmed the presence of the virus in two of 22 lung samples collected from necropsied animals that had died from respiratory disease. Additionally, FeMV antigen was demonstrated in the kidney, lymphoid, and brain tissues of two fully necropsied dogs [38]. The role of co-infection with other respiratory viruses and FeMV has to be further investigated. These data suggest that FeMV could be a significant canine respiratory pathogen.

The host spectrum of FeMV appears to also include noncarnivore species. Indeed, in Brazil, FeMV RNA was detected in a synanthropic marsupial, the white-eared opossum (*Didelphis albiventris*), and an FeMV strain from an opossum was isolated in Crandell Rees feline kidney lineage cells [39]. On phylogenetic analysis, the FeMV opossum strain clustered with FeMV-GT1 but formed a new branch [39].

It is clear that the host spectrum and tropism of FeMV go beyond the domestic cat and the kidney. This broader host range is also seen with CDV [40]. Morbillivirus host range and virulence are believed to be multifactorial, and the mechanisms involved are not completely understood. Generally, it is the viral proteins that interfere with the nonadaptive immune response of a host species that are responsible for virulence [40]. A major implication of a wide host spectrum for FeMV is the potential for interspecies transmission. This requires further investigation, as it may be that the susceptibility of dogs to FeMV allows for transmission between cats and dogs.

## 3. Epidemiology

Many studies have evaluated the prevalence of FeMV RNA in samples collected from live and necropsied cats following the initial report documenting FeMV in stray cats from Hong Kong in 2012 [11] (Table 1). It is not easy to compare the reported geographical prevalences as varied analytical methods have been used in the studies and the populations tested have differed with respect to their demographic characteristics, husbandry, lifestyle, and health status. However, higher prevalence of urinary FeMV RT-PCR positivity is found in older cats [41], in male cats compared to females [19], and in entire compared to neutered males [42]. A very high (52.9%) urinary RT-PCR positivity was found in cats from a cat shelter [28], although another study reported a higher prevalence in pets compared to shelter cats [19]. The urine of cats from suburban and rural areas were more frequently FeMV RT-PCR-positive than those from urban areas [41], and cats with outdoor access more frequently tested FeMV RT-PCR-positive than indoor cats [24,41]. Similarly, a higher prevalence was detected in cats in stray colonies compared to owned household cats [43]. Foundling cats and cats living in rescue catteries more frequently tested positive than nonfoundling cats and owned multi-cat household cats, respectively, in another study [41]. It is difficult to explain the differences seen with demographic data; however, male and entire cats usually have more aggressive interactions and may be a higher risk for infections transmitted by bites and mating. Similarly, outdoor access favours cat-to-cat interactions and contact with soil potentially contaminated with infected FeMV urine. In the case of shelters and rescue catteries, intra-species interactions depend on the management of facilities. When cats live indoors in a multi-cat environment, in addition to the risk of direct transmission of many infections, susceptibility to disease is generally increased by the chronic stress status of cats that can favour viral replication and shedding [44].

Urine and kidney tissues have been the sample types most often studied, with the aim of studying associations between FeMV RT-PCR positivity and kidney disease. However, wide ranges of urinary (range: 0.8–50.8%) and kidney (range: 7.4–80.0%) RT-PCR positivity have been detected in both healthy and sick cats (see Table 1 and Section 6).

Quantitative data on FeMV viral loads in positive samples from naturally infected cats are scarce; the Woo et al. (2012) study [11] reported only an overall median viral load of 3.9 × 10^3^ mL^−1^ RNA copies (range: 0.037–1.400 × 10^6^) on urine and rectal and oral swab samples. The FeMV urinary RNA viral loads in 40 cats were reported in a study aimed at developing a quantitative RT-PCR for detecting FeMV in biological specimens [50]; viral loads ranged from 2.98 × 10^3^ to 1.14 × 10^1^/μL [50]. No studies have compared viral loads from healthy and sick cats; nor have they compared samples obtained from normal and diseased tissues in infected cats.

The antibody prevalence of FeMV has been investigated in cats from many countries (Table 2).

Similar to RT-PCR positivity, antibody positivity prevalences vary widely (8.0–54.0%), and again, this could be due to the differing characteristics of the tested population and analytical methods used. The presence of serum antibodies against recombinant viral N protein has been investigated by Western blot [11,47] and against recombinant viral P protein by an enzyme-linked immunosorbent assay (ELISA) [51]. Additionally, high levels of antibodies against FeMV F protein were also detected by immunofluorescence assay (IFA) [27]. Different patterns of antibody reactivity against FeMV proteins have also been seen when feline sera were tested by whole-virus immunoblot analysis [45]. Thus, tests evaluating antibodies against single proteins might underestimate antibody prevalence.

The most widely used antibody testing technique is the IFA, which allows the detection of antibodies against all viral proteins [33,41,42,51,52,53,54]. However, genotype-specific IFAs are needed to evaluate the exposure of cats to specific genotypes. An IFA using two different cell lines infected with FeMV-GT1 and FeMV-GT2, respectively, was developed and validated for detecting antibodies against the two FeMV genotypes [52]. Cross-reactivity with CDV for FeMV-positive cat sera was excluded [52]. A high antibody prevalence (63.4%; 71/112) was detected in adult free-roaming rural cats from central to southern Chile, and 30% of cats had antibodies against both GT1 and GT2 [52]. Antibodies directed against only FeMV-GT2 were more prevalent in male cats, but only 10 FeMV-GT2-positive cats were found [52]. The same two genotype-specific IFAs were then used in a large retrospective study [53]. The authors tested 840 serum samples from 380 cats admitted to a veterinary teaching hospital in Germany with different diseases (43.0% of them for urinary disease) [53]. Similar to the study conducted in Chile, a high antibody prevalence (45%) was found, with 26.0% of cats being FeMV-GT1 antibody positive, 8.0% FeMV-GT2 antibody positive, and 15.0% positive for both genotypes. In this study, sex was not correlated to FeMV antibody status, and cats aged 3–4 years old were more likely to be antibody positive than older animals. Interestingly, pedigree cats were more frequently antibody positive and FeMV-GT1 antibody-positive compared to domestic shorthair cats.

Virus neutralising (VN) antibodies were not measured in epidemiological studies but evaluated in experimental investigation and in a case report [32,35].

Limited data regarding co-infection of FeMV with other feline pathogens exist. Feline immunodeficiency virus (FIV) antibody positivity was higher in cats shedding FeMV RNA in urine than in FIV-antibody-negative cats, and in another study, FeMV positivity was positively associated with both FIV and feline leukaemia virus (FeLV) infections [28,41]. In five FeMV-positive cat carcasses, tissue samples were positive for FeLV RNA (three cats), feline panleukopenia virus (four cats), feline coronavirus (FCoV) (one cat), and *Leishmania* spp. (one cat) [33]. Other infectious disease agents were detected in the cats of one large epidemiological study, but similar overall frequencies of these agents occurred in both FeMV-positive and FeMV-negative cats and between FeMV-positive genotypes (i.e., GT1, GT2, and both GT1 and GT2) [53].

Increased creatinine values were more commonly found in FeMV-positive cats compared to FeMV-negative cats, and increased creatinine was particularly associated with co-infection with both the GT1 and GT2 genotypes and with high FeMV-GT2 antibody titers [53]. In this study, five different diagnoses were found in the cats admitted with urological syndromes, namely urolithiasis, neoplasia, CKD, acute kidney injury (AKI), and feline lower urinary tract disease (FLUTD). A significant association occurred only between FLUTD and FeMV antibody positivity, the latter being either FeMV-GT1 antibodies or antibodies against both genotypes [53].

## 4. Acute FeMV Infection as Defined in Cats by Experimental Models and Sparse Field Data

Information from experimental infection studies has a low level of evidence based (EB) level (EB level III) [55], particularly when experimental infection does not mimic the natural infection route. Additionally, in vitro cultivation of the inoculated pathogens could attenuate their virulence. However, in vivo investigations can still provide useful information regarding the sequence of events following infection, particularly during early infection. Two different experimental infections with FeMV-GT1 and with FeMV-GT2, respectively, have provided feline models of FeMV acute infection [34,35].

The first experimental infection study delivered 10^4.6^ TCID50 of FeMV-GT2 Gordon strain intravenously to three groups of five young adult specific pathogen-free (SPF) cats [35]. Each group was sampled at different times for clinicopathological monitoring, detection of viraemia, viral excretion in urine and in nasal swabs (by RT-qPCR), and euthanised after 14, 24, and 56 days, respectively. A mild fever on days 3–5 post-infection (pi) was the only clinical sign observed in some cats. The complete blood counts (CBCs) and biochemical profiles were performed at different times in each group, providing data between day 14 and day 56 pi. Apart from a mild and transient leukocytosis, detected between days 20 and 49 pi, no other changes in CBC were found. However, one individual cat was severely leukopenic based on the graph reporting leukocyte counts on day 56 pi. Unfortunately, data relating to differential counts and/or from single individual cats were not provided, and so, any occurrence of the lymphopenia that is often observed in morbillivirus infections and that has been reported in experimentally infected cats [34] could not be determined. A sporadic increase in aspartate aminotransferase (AST) was reported in six cats from day 20 pi onwards, but the severity was not described.

The combination of different times of sampling in the three groups [35] provided information from day 1 to 56 pi for viraemia, from day 7 to 56 pi for urinary excretion, and from day 3 to 14 pi for nasal RNA detection. Viraemia was detectable with high viral loads in a variable number of cats in each group from day 1 to 49 pi (at around 10^3.5^ RNA copies/mL), with all cats tested on day 3 and day 5 found to be RT-quantitative (q) PCR positive. Peak viraemia (greater than 10^4^ RNA copies/mL) was associated with mild fever but was not associated with changes in cat behaviour. These findings suggest that the acute phase following natural FeMV infection is unlikely to be detected by veterinarians, as owners would rarely have a reason to seek veterinary help for their animals.

Urinary excretion was confirmed in some cats at all time points, with all tested cats found positive from day 20 to day 56 pi. Urinary viral loads were higher than in the blood from day 20 pi onwards, with concentrations of RNA copies greater than 10^4^/mL up to day 49 pi. These findings confirm the field observations that have found lower percentages of positive blood samples compared to urine samples [11,17,41,50] or, indeed, an absence of positive blood sample test results [19,33]. The duration of viraemia appeared to be variable, as suspected under natural conditions [41]. Nasal swabs tested positive in some cats when viraemia peaked, but viral loads were mostly less than 10^3^ RNA copies per swab [35]. Antibody seroconversion was detected early and was associated with a declining viral load in blood; however, viraemia was observed up to day 49 pi, in line with the frequent occurrence of naturally infected cats testing serum antibody positive, as well as RT-PCR positive in urine [32,41].

Although necropsies did not reveal gross lesions at any time point of euthanasia, histopathological examination of kidney, liver, and spleen samples were performed, as well as immunohistopathology on kidney samples [35]. Histopathological lesions in the kidney and liver and diffuse activation of lymphoid follicles in the spleen were reported at all time points [35]. Renal tubular abnormalities were observed in all cats and immunohistology-localised FeMV nucleoprotein at the apical surface of epithelial cells from the renal cortex. Multifocal tubular (hyaline or granular) casts were observed from day 14 pi with all cats affected later on, sometimes with degeneration of the lining of the epithelial tubular cells. Subsequently, multifocal tubular mineralisation and multifocal chronic TIN were found in some cats, and lesions observed on day 56 were in general considered more prominent. The above findings suggest that, under natural conditions, urinalysis with microscopic detection of urinary casts could provide early information regarding the acute tubular damage that was detected in this experimental study, as well as the detection of a tubular pattern of proteinuria detectable by urine protein electrophoresis (UPE). Indeed, Crisi et al. (2020) [56] found a tubular pattern frequently in a retrospective urinary sodium-dodecyl-sulphate-polyacrylamide gel electrophoresis (SDS-PAGE) evaluation of FeMV-positive cats [56]. Hepatic lesions were reported as mild, and these lesions were found in all cats at all time points despite only sporadic AST increases occurring pi [35]. Multifocal lymphoplasmacytic portal and interstitial hepatitis and hydropic degeneration of hepatocytes were detected. Most cats also showed multifocal acute portal haemorrhages, and at day 56 pi, portal biliary proliferation and fibrosis were observed in one cat.

This experimental model evidenced urinary viral loads similar (>10^4^ RNA copies/mL) to those observed in a healthy cat and in one cat with CKD naturally infected with FeMV-GT1 [27,50]. Moreover, the occurrence of FeMV antigen in epithelial tubular cells, typical of natural infection, was confirmed [11,41,50,57,58]. The low and transient viral loads observed in the nasal mucosa at peak viraemia, with no signs of upper respiratory tract disease, suggest that the risk of transmission via the respiratory route is low. Similarly, Woo et al. (2012) [11] did not find RT-PCR-positive nasal swabs in any of the 457 stray cats studied, while 53 of 457 urine samples tested positive. At present, scant information is available on mucosal FeMV positivity rates, and one field study found only 1/16 cats sampled by oral and rectal swabbing was positive in both samples (viral loads were not measured). Rectal swabs (4/457) were less frequently positive compared to the urine samples (53/457) in stray cats [11]. Intestinal samples of an FeMV-positive cat with severe necrotising enteritis tested RT-PCR positive, but IHC evaluation of the intestine was negative [58]. Unfortunately, clinicopathological evaluation earlier than two weeks pi, including early markers of renal dysfunction, were not performed in this experimental study.

Interestingly, the acute phase of infection was clinically irrelevant with no overt cat clinical signs nor gross lesions at necropsy, but most of these cats had already suffered from multifocal chronic TIN and mild hepatic lesions. TIN is found in natural feline FeMV infection and is also the most common histopathological pattern of cat CKD with causative agents mostly undetected [59]. Overall, these experimental data confirm a role for FeMV as a potential causative agent of feline TIN [35]. Importantly, hepatic lesions were similar to those reported in naturally infected cats with positive FeMV antigen IHC in hepatocytes and monocytes, but in this field study, some cats were also FCoV positive [24]. A role for FeMV in cases of lymphoplasmacytic portal hepatitis merits further evaluation.

The second experimental model of feline FeMV infection focused on the early spread of the virus after airway transmission, which may be an infection route in natural infection (Figure 1).

Two groups of three young domestic shorthair male cats (16–17 weeks old) were infected by intratracheal (10^6^ TCID50) and intranasal (2 × 10^5^ TCID50) routes with two recombinant viruses of an FeMV-GT1 unpassaged strain obtained from a chronically infected cat [34]. Of note, a fluorescent protein-expressing recombinant FeMV was used to track virus spread during necropsy and to identify infected cells. In the first group, individual cats were sampled (blood, urine, and throat and nasal swabs) at different time points and euthanised at days 7, 14, and 28 pi. The second group of cats was sampled on days 2, 5, 6, and 7 pi, when all cats were euthanised to study the peak of early acute infection. Increased temperatures occurred at around day 5 pi in both groups with lymphopenia peaking at the same time. The cats then underwent progressive recovery. Viraemia was confirmed by flow cytometry in white blood cells (WBCs) on days 6–10 pi and in lymph nodes and bronchoalveolar lavages (BAL) on day 7 pi. Virus was isolated from WBCs on days 4 to 14 pi, from urine on days 12 to 28 pi, and from lung samples on day 7 pi only and was never isolated from throat or nose swabs. All six studied cats were necropsied and post-mortem macroscopic bioimaging evaluation confirmed virus lymphotropism during the early acute phase (day 7 pi), when all lymph nodes, thymus, and tonsils were highly positive. IHC in lung and lymph node tissues showed that most of the infected cells were monocytes/macrophages. Data from necropsies showed that the peak of virus detection in the respiratory tract was at day 7 pi, when it was detected in the BAL, the lung interstitium, bronchial tissue, and the bronchial-associated lymphoid tissue. Urinary FeMV isolation and the presence of renal lymphoplasmacytic lesions occurred later (days 14 and 28 pi) and was associated with positive IHC in the medullary tubular epithelium of kidney sections on day 28 pi. This FeMV-GT1 experimental model showed hallmarks typical of early morbillivirus infections in the infected cats, such as lymphotropism with viral detection in WBCs and lymphopenia [34].

The clinical evaluation of experimentally infected cats showed that acute FeMV disease is not characterised by overt clinical signs despite fever and urinary and pathological demonstrations of AKI and the occurrence of lymphopenia [34,35]. However, this could be different in noncontrolled situations where host and virus variables could lead to different outcomes. This gap of knowledge has been investigated by Ito et al. (2023) [60] after the detection of FeMV in a cat that was subjected to an investigation of unknown viruses as part of a study in cats with fever using unbiased next-generation sequencing [61]. The retrospective controlled study was performed in an area endemic for the zoonotic severe fever with thrombocytopenia syndrome (SFTS) caused in East Asia by Huaiyangshanbanyang virus; most of the studied cats had fever, leukopenia, thrombocytopenia, and jaundice, but SFTS and parvovirus infections had been excluded by blood PCR [61]. FeMV-GT1 (mostly of subtype A) was detected in 32 of 102 plasma samples (31.4%) by RT-qPCR (threshold cycle (Ct) value range: 27.4–39.0), and FeMV RNA was never found in 374 control samples from sick cats [61]. All positive cats but one presented with lethargy and anorexia, and fever was observed in about half of cases. Half of the cases also showed jaundice. Around 50% of positive cats were thrombocytopenic, with most of these thrombocytopenic cats having fever. Leukopenia (WBCs lower than 2.9 × 10^3^/μL) was found in seven cats, but differential counts were unfortunately not reported. Other causes of feline fever (apart from SFTS and parvoviral infections) were not specifically investigated. Retroviral (FIV and FeLV) co-infections affecting feline immunocompetence were evaluated in only 13 cats, with three being FIV positive. The duration of clinical signs was unfortunately not known in this case series, but the data supported a role for FeMV in the clinical status of cats with acute fever.

With respect to the post-mortem findings in cats suspected of acute FeMV disease, three case reports are available and are worthy of description [58,60]. Various samples tested FeMV RT-PCR positive in a cat that had died about four days after the onset of lethargy and anorexia [60]. Necropsy did not reveal the cause of death, but high viral loads in the spleen (Ct 17.7), lung (Ct 22.1), liver (Ct 23.3), urine (Ct 23.6), blood (Ct 24.4), rectal swab (Ct 24.5), kidney (Ct 25.3), oral swab (Ct 29.0), and lymph nodes (Ct 34.5) were detected. Unfortunately, FeMV IHC was only performed on the kidney, spleen, and lymph node samples. Immunopositivity was found in renal tubular cells, in macrophage and lymphocyte infiltrates surrounding positive tubules, in lymphoid follicles in the mandibular lymph node, and in macrophages and lymphocytes in splenic lymphoid follicles [60]. The role of FeMV in the death of the cat was not proven, but the systemic viral dissemination was suggestive of acute infection [60]. Viral systemic spread of FeMV was detected in another two cats that died with acute multifocal necrotising haemorrhagic cystitis associated with two different bacterial infections (*Escherichia coli* in one case and *Pseudomonas aeruginosa* in the other) and suspected septicaemia, detected at necropsy [58]. Diffuse renal tubular vacuolation with mild or moderate segmental multifocal membranous glomerulopathy were observed, as well as viral inclusion bodies in tubular epithelial cells [58]. High viral loads were measured by RT-qPCR in the urine (Ct 24.8) and FeMV-GT1 infection was diagnosed in both cats. Interestingly, kidney infection was confirmed by RT-qPCR (Ct 34.2) and IHC evaluation, but no associated inflammatory infiltrates were observed within the renal tissue. Lung viral load was lower (Ct 37.4) than urinary bladder and small intestine loads (Ct 34.8), and the cytoplasm of various epithelial cells (transitional, tracheal, bronchial, and bronchiolar), macrophages, and lymphoid cells in the spleen and lymph node occasionally tested IHC positive. Liver and brain samples were also RT-PCR positive, and astroglia and oligodendroglia cells were FeMV positive by IHC. Epitheliotropism (particularly in renal tissue), lymphotropism, and neurotropism of FeMV were observed in these two young cats, but a role in their death was not evident [58]. Data from these necropsied cats confirmed that systemic spread of FeMV occurs.

## 5. Persistent Infection Documented in Natural Feline Infection Studies

Feline models of chronic FeMV infection are not available. The high percentages of RT-PCR-positive cats detected worldwide since the discovery of FeMV (Table 1) are supportive of a chronic course of infection. Some positive cats have been followed up longitudinally to monitor changes in their health status and the duration of urinary excretion [27,32,33,41,50], as outlined below.

Longitudinal field studies are reliant on owner compliance for sampling, and therefore, the information available is usually scarce and fragmented. The tendency for FeMV to persist in vivo was repeatedly reported in both healthy cats and cats with CKD. Urine samples of a healthy adult cat were found to be PCR positive 15 months after the first detection of FeMV-GT1 RNA in urine [27]. FeMV RNA was amplified and sequencing of the haemagglutinin H gene performed, which showed that the sequences were identical in the two urine samples; their viral loads were also similar at 9.8 × 10^4^/mL and 7.8 × 10^4^/mL, respectively [27]. A 15-year-old cat with CKD shed FeMV-GT1 in its urine with viral loads ranging from 3.69 × 10^2^ to 1.03 × 10^1^ copies of RNA/μL in 33/42 samples tested from diagnosis until death 110 days later [50]. The Ct values and sequences of FeMV-GT1 in the urine samples of five cats were unchanged during an epidemiological study that lasted 8–10 months [33]. Similarly, long-term shedding of FeMV-GT2 was observed in two cats [32]. The duration of shedding in the urine of a 13-year-old cat with CKD was documented for six months and for two years in a 6-year-old animal with FLUTD. Both cats had neutralising antibodies (with a titer greater than 256) against FeMV-GT2 [32]. Among 13 of 27 cats followed up by Donato et al. (2021) [41], various patterns of FeMV-GT1 antibodies and RT-PCR positivity in the blood and urine were observed over time. Chronic urinary shedding was a frequent event, as eight cats tested positive in multiple urine samples and up to 360 days after first detection, with two of them having no direct contact with other cats. Interestingly, antibody seroconversion did not occur in all cases, as four cats positive in urine samples for 21–360 days did not develop serum antibodies during the monitoring period. Finally, infection could be cleared spontaneously over time, as one cat that was followed up for six months converted to an antibody-negative status associated with the cessation of FeMV RNA urinary shedding [41].

All these data support the hypothesis that both FeMV genotypes are excreted for a long time in the urine in stable concentrations and that this may occur also in seropositive cats with VN antibodies. Prolonged CDV urinary shedding in dogs with VN antibodies has been reported, representing an insidious source of infection to other dogs [62,63]. Subacute and chronic CDV infections can also trigger in dogs an immunopathological process resulting in demyelinating leukoencephalitis in dogs [64].

## 6. Difficulties in Obtaining Field Study Evidence of FeMV Pathogenicity

Since the discovery of FeMV in 2012 [11], in the renal tubular cells and lymph nodes in two stray cats affected by TIN, research studies have focused on the role of FeMV in feline kidney pathology and in CKD. This 2012 study also included a case-controlled prospective investigation that provided evidence for a significant association between FeMV infection and TIN (EB level I) [11,55]. Thereafter, kidney tissues have been often studied with a higher percentage of positive samples found (ranging from 7.4 to 80.0%) compared to other tissues and urine [11,19,24,33,42,48,50]. Kidney histological and IHC evaluation, performed in some studies, aimed to detect associations between any pathological changes, the detection of FeMV, and the occurrence of FeMV in any lesions [11,33,50,57]. However, field studies had used different approaches and given contradictory results. As in the first study [11], FeMV detection can be associated with tubular damage and the presence of inflammatory infiltrates and intralesional FeMV detected by IHC [11,24,42,57]. The tissue injury scores of tubular lesions were higher in FeMV-positive tubular sections, as well the severity scores of glomerulosclerosis and capillary thickness [57]. Conversely, in other studies, the lesions observed were similar to those detected in FeMV-negative cats [24,33]. This does not exclude a role for FeMV in the pathological changes observed, as the virus could have been cleared from tissues prior to examination. It should be remembered that TIN is the most common diagnosis in feline kidney pathology and the most common cause of feline CKD [26]. When tubular injury involves the basement membrane (tubulorrhexis), inflammation spreads to the interstitium, and focal TIN occurs [65]. Feline tubular cells typically accumulate lipids in the cytoplasm, and leakage of lipids into the interstitium in the case of tubulorrhexis enhances the inflammatory response [66]. Interestingly, a few weeks after experimental infection with FeMV-GT2, multifocal chronic TIN was observed in some cats, similar to what had been seen in feline experimental models of renal ischaemia [35,66]. Indeed, AKI-to-CKD transition is usually triggered by hypoxia regardless of the cause of AKI [65]. In contrast, in dogs, immune-complex glomerulopathies are the most common primary chronic kidney lesions detected, and secondary tubular damage can follow, and dogs with terminal distemper may also have proteinuria with immune-complex glomerular disease and tubular lesions documented by histopathology [67].

A few clinical surveys [22,41,51] have demonstrated an association between FeMV RT-PCR-positivity and CKD, other urinary pathologies, or increased serum values of creatinine. However, no associations have been found in many other investigations [19,20,24,26,28,33,47,48,49]. A limitation of some studies is due to the criteria used to select CKD and control cases, and in some reports, only a low number of cats have been tested. Moreover, in cross-sectional studies detecting viral RNA in urine samples, there is a risk of negative RT-PCR results despite infection because of possible intermittent urinary shedding. Additionally, CKD can be clinically diagnosed months or years after a pathogen has triggered the pathologic process leading to the development of TIN, and in the meantime, the active infection might have resolved. Finally, cats examined in field studies can be exposed to a wide range of infectious and noninfectious causes of CKD [68], and these confounding factors are not easily investigated and/or recognised. For example, Crisi et al. (2020) [56] compared the clinical, haematological, and urinary parameters in cats with positive urine FeMV RNA RT-PCR results, cats with CKD, and healthy cats [56]. No cats in the CKD group tested FeMV positive; however, some degree of early renal damage, less severe than in the CKD group cats, was demonstrated in those cats testing FeMV RT-PCR positive in urine. Of note, this study performed UPE as well, and FeMV-positive cats showed a frequent tubular pattern of proteinuria, and the three necropsied cats were diagnosed with TIN [56]. This clinical study showed similarities, with the results obtained in the experimental infections described above showing early kidney damage caused by FeMV [34,35,56]. Interestingly, a transient proteinuria and cylindruria were documented in the FeMV RNA-positive urine of a cat diagnosed and treated for cholangiohepatitis [48].

Fewer data on pathology are available for infected tissues other than kidney. Liver involvement is definitely of high clinical relevance. Lymphocytic cholangiohepatitis is a common inflammatory hepatobiliary disease in cats histologically characterised by lymphocyte infiltration in the portal region with various degrees of fibrosis and bile duct proliferation [69]. It is often subclinical with variable biochemical parameter abnormalities, and the causes triggering the aberrant inflammatory process are unknown [69]. Sometimes, liver FeMV positivity (by RT-PCR and/or IHC) was associated with kidney positivity in cats affected by diffuse cholangiohepatitis, as seen in experimental infection [24,35]. Data on spleen, lymph node, lungs, intestine, urinary bladder, infections, and associated pathology are very sparse in field studies and case reports [33,58], and further investigation is needed.

Concerning neurotropism and neurovirulence, neurologic signs have never been reported in experimental nor natural feline infections [34,35]. Experimental studies have not investigated the CNS of infected cats, and FeMV neurotropism seems to be associated with less extensive viral replication compared to epithelial and inflammatory cells [33,34,35]. However, brain glial cells of both cats and dogs and dog neurons were found to be FeMV positive by IHC in natural disease [38,58]. Thus, FeMV neurovirulence seems to be low compared to other morbilliviruses that more frequently cause encephalitis in animal hosts (CDV, PDV, and CeMV) [40]. Cellular receptors favouring the neurovirulence of CDV in dogs are not known, and SLAM expression is very low in the CNS [64].

FeMV infection (confirmed by both RT-PCR and IHC) of tubular cells without inflammatory reactions has been seen in some cases [41,58], and factors promoting inflammation and associated damage to tissues (primarily in the kidney and liver) remain unknown.

It has to be considered that TIN and lymphocytic cholangiohepatitis are common feline kidney and liver pathologies, respectively, that occur worldwide; FeMV is just one possible cause [59,69].

## 7. Conclusions and Future Directions

The limited number of experimental investigations and difficulties in obtaining robust data from field studies currently leave important questions to be addressed with further studies of FeMV infection and associated diseases. FeMV was characterised as a morbillivirus when it was discovered in 2012 [11], but based on phylogenetic analysis of the H protein amino acid sequences, it diverges from the six classical morbilliviruses [15,34]. Feline cell receptors other than SLAMF1 and nectin-1 might bind FeMV H glycoprotein in tissues, resulting in severe viral infection. Moreover, host-dependent factors can modulate morbillivirus virulence; for example, the reduced cathepsin availability in feline lymphocytes has been considered responsible for the less severe lymphodepletion in cats compared to that observed in the CDV model in ferrets [34]. Other mechanisms may affect FeMV respiratory pathogenicity (currently observed only in dogs) [38] and neurovirulence. Cellular receptors favouring neurovirulence of CDV in dogs have not been identified, and SLAM expression is very low in the CNS [64]. The importance of the nonadaptive immune response should also be taken into consideration [40].

Spontaneous transmission roues and possible interspecies transmission must be proven. The respiratory route is the most probable, and urine is likely an important source of infectious virus as FeMV is frequently (Table 1) and chronically [32,33,41] shed in urine, although there are no data about the infectiousness of FeMV excreted in urine. However, when considering the behavioural importance of both olfactory exploration of cats for detecting scent marks and the release of scent when they spray urine, it is easy to understand why a virus shed chronically in the urine with possibly a low rate of acute lethality is a good candidate for endemic infections in feline populations [70]. Moreover, cat reciprocal facial rubbing and allogrooming behaviour [71] would favour transmission of infectious virus shed by mouth and nose, beyond shared bowls and litter trays.

Regarding FeMV pathogenicity, a wide spectrum of clinical outcomes appears to be possible, from subclinical infections to acute and/or chronic disease and lethal outcomes. Interactions among viral and host factors are probably involved. These may include viral genetic diversity, co-infections (with both FeMV genotypes and other feline pathogens), and co-morbidities, particularly those impairing cat immunocompetence. The proportion of cats that develop long-term life-threatening renal disease after FeMV infection is unknown, as are the risk factors that drive a poor prognosis for kidney function. It is possible that cats can clear FeMV infection, but this has to be proven by extensive longitudinal studies, as well as the duration of immunity after clearing the virus [41]. A disease associated with acute infection can develop and clinicians should be aware of the possibility that FeMV causes AKI when clinicopathological abnormalities suggest this diagnosis and no other cause is found [34,35,56]. Similarly, FeMV infection should be considered in cases of acute febrile syndrome of unknown origin [60]. In these cases, acute FeMV infection can be diagnosed by RT-PCR in blood and also by IHC in cats that have died following acute disease [34,35,58,60]. Antibody detection (by IFA detecting both GT1 and GT2 strains) supports acute disease diagnosis when seroconversion is evidenced, as seen in experimental models [34,35].

As in all cases of AKI, CKD is a possible progressive sequelum that has to be clinically monitored and managed in cats [68]. It is recognised that CKD can be clinically diagnosed months or even years after a pathogen triggered the pathological process that led to the development of TIN. Since the infection could have resolved in the meantime, based on current knowledge, ABCD does not recommend routine screening of FeMV infection in healthy cats, nor in those with CKD.

In conclusion, it is difficult to define the clinical relevance of infections caused by microorganisms that are endemic in their host species, as is the case for FeMV. There are other important examples in feline medicine, such as FCoV, *Bartonella* spp., and *Toxoplasma gondii* infections. FeMV seems to be an additional “evil” in this Pandora’s box. Hopefully, the combination of data from future experimental and longitudinal field studies will progressively increase our understanding of the aetiological role of FeMV in feline diseases, including CKD.

## Figures and Tables

**Figure 1 viruses-15-02087-f001:**
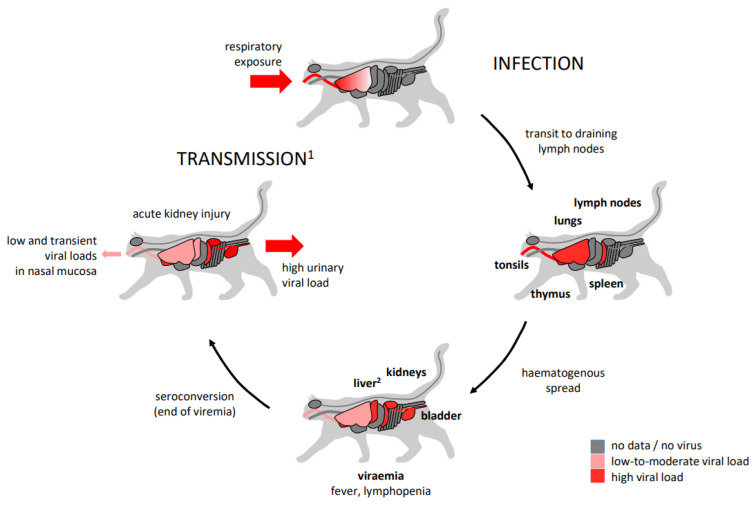
Feline model of acute feline morbillivirus (FeMV) respiratory infection with FeMV-GT1 [34] and intravenous infection with FeMV-GT2 [35]. The main phases of the first weeks of infection are summarised with a focus on tissues infected and viraemia, clinical signs, seroconversion, acute kidney injury, and viral shedding. ^1^: spontaneous transmission has not been studied. ^2^: presence of virus in the liver is reported only in [35].

**Table 1 viruses-15-02087-t001:** Worldwide FeMV prevalence data from cats reported in chronological order and in relation to the country studied, the characteristics of the cat population sampled, and the sample types tested by reverse transcriptase–polymerase chain reaction (RT-PCR).

Country (Year)	Cat Population	Number of Cats Sampled	Sample/Tissue (% of Positive RT-PCR)	Overall % of Positive RT-PCR ^(a)^	Reference
Hong Kong (2012)	Strays	457	Urine (11.6)	12.3	[11]
			Blood (0.2)		
			Rectal swabs (0.8)		
Mainland China (2012)		16	Oral swabs (6.2)	6.2	
			Rectal swabs (6.2)		
Japan (2014)	Admitted to clinics	82	Urine (6.1)	n. r.	[17]
		10	Blood (10.0)		
		10	Kidney (40.0)		
Japan (2014)	Client-owned	13	Urine (23.1)	n. e.	[45]
Japan (2016)	Admitted to clinics	166	Urine (15.1)	n. e.	[46]
USA (2016)	n. r.	327	Urine (3.0)	n. e.	[27]
Japan (2016)	Strays/client-owned	100	Urine (17.0)	22.0	[42]
			Kidney (18.0)		
Brazil (2017)	Multi-cat household °	17	Urine (52.9)	n. e.	[28]
	Client-owned	35	Urine (8.6)		
Turkey (2017)	Client-owned	96	Urine (3.1)	5.4	[24]
		15	Kidney (26.0)		
			Lymph nodes (13.0)		
			Lung (6.0)		
			Spleen (6.0)		
			Intestine (6.0)		
			Liver (6.0)		
UK (2018)	Client-owned geriatric	40	Urine (12.5)	n.e.	[47]
Italy (2019)	Strays	6	Urine (16.7)	3.2	[48]
	Client-owned	59	Urine (0.0)		
	n. r.	27	Kidney (7.4)		
Malaysia (2019)	Sheltered °/client-owned	124	Urine (50.8)	39.4	[19]
		93	Blood (0.0)		
		25	Kidney (80.0)		
Germany (2019)	n. r.	723	Urine (0.83)	n.e.	[32]
Italy (2020)	Colony	69	Urine (31.8)	n.e.	[33]
	Client-owned	127	Urine (8.6)		
	Colony	7	Kidney (57.1)	22.8	
			Urinary bladder (14.2)		
			Spleen (28.5)		
			Lymph nodes (14.2)		
	Client-owned	28	Kidney (10.7)		
			Urinary bladder (10.7)		
			Spleen (3.5)		
			Brain (3.5)		
Mainland China (2020)	n. r.	64	Urine (9.3)	n.e.	[21]
Thailand (2020)	Sheltered °*	31	Urine (19.3)	11.9 (Sheltered: 29.5; Client-owned: 6.5)	[20]
	Client-owned ^§^	100	Urine (13)		
	Sheltered °*	61	Blood (19.6)		
	Client-owned ^§^	100	Blood (0.0)		
Brazil (2021)	Client-owned	56	Urine (26.7)	n.e.	[49]
	Multi-cat household	82	Urine (28.0)		
	Sheltered	138	Urine (42.0)		
	Total	276	Urine (34.7)		
Italy (2021)	Client-owned	127	Urine (3.9)	n.e.	[26]
	Cattery	23 ^&^	Urine (26)		
	Total	150	Urine (7.3)		
	Client-owned	40	Kidney (7.5)	n.e.	
	Cattery	10	Kidney (10.0)		
	Total	50	Kidney (8.0)		
Italy (2021)	Outdoors	111	Urine (18.9)		[41]
	Indoors	106	Urine (14.2)		
	Total	223	Urine (16.1)		
	Outdoors	111	Blood (2.7)	18.5	
	Indoors	100	Blood (2.0)		
	Total	211	Blood (2.4)		
	Indoors/Outdoors	10	Kidney (10.0)		
			Urinary bladder (10.0)	10.0	
			Mandibular lymph nodes (10.0)		

(a): % of cats with at least one tested sample/tissue found RT-PCR positive; °: cats from a unique facility; ^§^: urine and blood samples were obtained from cats from different households; *: urine samples were obtained from 31/61 shelter cats that had blood samples tested; ^&^: three pools of urine each from 10 additional cats housed together in the same cattery were also tested and two tested positive; n. r.: not reported; n. e.: not able to evaluate.

**Table 2 viruses-15-02087-t002:** Worldwide anti-FeMV antibody prevalence in cats reported in chronological order and in relation to the country studied, the characteristics of the cat population sampled, and the serological technique used.

Country (Year)	Cat Population	Number of Tested Cats	Prevalence (%)	Assay	Reference
China (2012)	Strays	457	27.8	WB	[11]
Japan (2014)	Client-owned	13	23.1	WB	[45]
Japan (2016)	Strays/client-owned	100	21.0	IFA	[42]
Japan (2017)	Not reported	100	22.0	ELISA	[51]
UK (2018)	Client-owned geriatric	72	31.0	WB	[47]
Italy (2020)	Colony	69	21.7	IFA	[33]
	Client-owned	127	17.3		
	Total	196	18.9		
Chile (2021)	Rural free-roaming	112	54.0 39.0	GT1-IFA * GT2-IFA *	[52]
Germany (2021)	Admitted to hospital	380	26.0 8.0	GT1-IFA ** GT2-IFA **	[53]
Italy (2021)	Outdoors	103	18.5	IFA	[41]
	Indoors	90	10.0		
	Indoors + Outdoors	193	14.5		

WB: Western blot; IFA: immunofluorescence assay; ELISA: enzyme-linked immunosorbent assay; *: sera tested with two different assays developed for genotypes 1 (GT1) and 2 (GT2) of FeMV; 30% of cats were antibody positive to both genotypes, and in total, anti-FeMV antibody prevalence was 63%; **: sera tested with two different assays developed for FeMV-GT1 and FeMV-GT2; 15% of cats were antibody positive to both genotypes, and in total, the anti-FeMV antibody prevalence was 49%.

## Data Availability

No new data were created or analysed in this study Data sharing is not applicable to this article.

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
