# Peer review of "Feline Morbillivirus: Clinical Relevance of a Widespread Endemic Viral Infection of Cats"

_viruses, 2023, doi:10.3390/v15102087_

Round 1

Reviewer 1 Report

This manuscript titled “Feline morbillivirus: clinical relevance of a widespread endemic viral infection of cats” aims to reviews the Clinical Relevance of Feline Morbillivirus. The manuscript is well-organized and has certain significance. It was addressed a specific gap in the field, the references are appropriate, However, there are some problems in this manuscript that need to be revised;

1 The language needs considerable attention.

2 There are too few references for the review.

3 The author can add some model pictures to better display.

Reviewer 2 Report

General and major comments

The article is of interest to the scientific community and summarizes the current knowledge on Feline Morbillivirus. It is overall clear and reads well. It however requires some major changes.

First, I am not convinced by the subdivision used in titles 4., 5. and 6. These titles are rather “methodological”. The corresponding sections include a highly descriptive and likely overdetailed retranscription of the content of original articles on the topic with little attempt at synthetizing the experimental and field observations into a comprehensive analysis of current knowledge and gaps on FeMV pathogenesis. I would recommend a more systematic and synthetic approach combining all available data to highlight the key features of FeMV infection in cats.

Next, a deeper comparison with well-know pathogenic Morbilliviruses, chiefly canine distemper virus (Morbillivirus canis), would have been welcome. In this respect, I strongly disagree with the conclusion at lines 541-543 and stating that “FeMV lymphotropism, epitheliotropism (particularly in kidney and urinary bladder), and neurotropism, typical of morbilliviruses, were observed in both natural [20,33] and experimental studies [34,35].”. This is incorrect and not supported by the content of the manuscript. Epitheliotropism is far less widespread than that of other Morbilliviruses, with little respiratory tropism, and neurotropism is minimal and does not result in any clinical signs. This sharply differs from CDV pathogenesis. Of note, there is no discussion in the manuscript about the virological mechanisms underlying this peculiar tissular tropism of FeMV. What role are SLAM receptors likely to play in this respect? Are other, yet unknown, receptors potentially involved in the epithelial tropism of Morbilliviruses?

Other important and comparative questions are: What about chronic infections associated with other Morbilliviruses in their hosts? How do the renal lesions observed in FeMV-infected cats compare with renal lesions reported for other Morbilliviruses in their respective hosts (since CDV has also been associated with renal lesions in dogs)?

Similarly, an up to date phylogenetic tree illustrating the genetic distances between the different members of the Morbillivirus genus would be appreciated. Feline Morbillivirus seems to have diverged very early from other Morbilliviruses, which likely accounts for its quite different pathogenesis.

Finally, although many RT-qPCR results are reported in the manuscript, not a single Ct value is mentioned. Since the emergence of molecular methods, it has become more and more obvious that many previously thought pathogenic viruses can actually be detected in healthy carriers, and are thus part of the “virome” of this particular host species. Yet, quantitative methods such as viral titration or (RT-)qPCR proved to be useful tools to discriminate between active/pathogenic infections (low Ct values, high viral loads) and asymptomatic infections (high Ct values, low viral loads), as again recently best exemplified by the SARS-CoV-2 pandemics. Since the potentially “pathogenic” nature of FeMV appears to be still a mater of debate, this information could be helpful in the current manuscript, in particular to link the renal lesions with associated viral loads in the renal parenchyma or in the urine.

Specific comments

Lines 38: please replace “tubule-interstitial nephritis” with “tubulointerstitial nephritis

Line 49: “acute renal damage” is likely not an accurate description of the renal lesions reported in the literature. Indeed, both “acute tubular injury” (also called “acute tubular necrosis”), referring to an active destruction of tubular epithelial cells with casts formation, and chronic lymphoplasmacytic tubulointerstitial nephritis were observed. This is a relatively unusual pattern. Many (mostly toxic, sometimes viral) etiologies inducing acute tubular injury cause “pure” acute lesions, while agents responsible for chronic TIN (Leptospira interrogans for instance) induce mostly chronic changes but minimal acute tubular necrosis. The coexistence of these lesions suggests ongoing tubular necrosis over a long period of time, resulting in significant acute and chronic changes, hence a chronic infection. There remains that some level of lymphoplasmacytic TIN is a classical observation in most feline kidneys at necropsy, raising questions about the possible link between this lesion and FeMV (as correctly mentioned and discussed later in the manuscript).

Lines 171-175 and Table 1: as mentioned in the general comments, trying to link the PCR detection rate with clinical signs is likely irrelevant for an endemic virus with such a high prevalence. Comparing viral loads between healthy and sick cats could give a better clue about the pathogenic potential of FeMV.

Tables 1 and 2: the number of decimals is mostly 1, but sometimes 2 or 0. Please use one decimal in all numbers.

Lines 388-390: this is particularly interesting. As a consequence, it appears that serological methods are likely to underestimate the actual exposition rate of cats to FeMV in the population. Next, could it be suggested that the humoral immunity to FeMV is short-lived? As far as I know the immunity in dogs vaccinated against CDV has been shown to be relatively long-lasting and well-correlated with antibody titers. This, again, might appear as a striking difference with CDV.

More generally about the various outcomes reported after natural and experimental infections by FeMV, it might be worth mentioning that additional studies are required to better address the potential role of coinfections (for instance FIV, FeLV or FeCoV) or other factors affecting the cat’s immunity (endocrine and other general conditions).  

Round 2

Reviewer 2 Report

I wish to thank the authors for the extensive changes implemented in the manuscript. My comments have overall been properly addressed and a good compromise has been proposed regarding the comparison with other Morbilliviruses. I have no further remarks.